# Exploring Sugary Drink Consumption and Perceptions among Primary-School-Aged Children and Parents in Australia

**DOI:** 10.3390/nu16193320

**Published:** 2024-09-30

**Authors:** Zenobia Talati, Jessica Charlesworth, Katlyn Mackenzie, Thomas McAlpine, Gael Myers, Caroline Miller, Liyuwork M. Dana, Moira O’Connor, Barbara A. Mullan, Helen G. Dixon

**Affiliations:** 1School of Population Health, Curtin University, Kent St, Bentley 6102, Australiakatlyn.mackenzie@student.curtin.edu.au (K.M.); thomas.mcalpine@curtin.edu.au (T.M.); liyuwork.dana@curtin.edu.au (L.M.D.); m.oconnor@curtin.edu.au (M.O.); barbara.mullan@curtin.edu.au (B.A.M.); 2The Kids Research Institute Australia, 15 Hospital Ave, Nedlands 6009, Australia; 3enAble Institute, Faculty of Health Sciences, Curtin University, Perth 6102, Australia; 4Cancer Council WA, 420 Bagot Rd, Subiaco 6008, Australia; 5South Australian Health and Medical Research Institute, Adelaide 5001, Australia; caroline.miller@sahmri.com; 6School of Public Health, University of Adelaide, Adelaide 5001, Australia; 7Cancer Council Victoria, 200 Victoria Parade, East Melbourne 3002, Australia; helen.dixon@cancervic.org.au; 8Melbourne School of Psychological Sciences, The University of Melbourne, Parkville 3052, Australia

**Keywords:** sugary drinks, sugar-sweetened beverages, child, parent

## Abstract

Background: Sugar-sweetened beverages (SSBs) account for a significant proportion of sugar in the diet of children and are directly associated with obesity in this group. While there have been many studies on adolescent SSB consumption, few studies have examined the predictors of SSB consumption in primary-school-aged children. The aim of this study was to understand the degree to which a child’s consumption across a range of beverages is influenced by their own attitudes and by their parents’ attitudes and parents’ consumption behaviours. Methods: A survey of 1611 Australian parent–child dyads asked children (aged 4–11) and their parents to rate a variety of drinks in terms of healthiness, taste, and cost and indicate the amount of these drinks consumed in a typical week. Zero-inflated regression analyses were conducted to identify the strength of association between children’s weekly beverage consumption, their perceptions of each beverage, their parents’ perceptions, and their parents’ weekly beverage consumption. Results: Parental consumption of a specific beverage was the strongest predictor of child consumption of that beverage, more so than the children’s perceptions of the beverage. Conclusions: These findings provide insights for developing parent and child education programmes to reduce SSB consumption.

## 1. Introduction

### 1.1. Sugar-Sweetened Beverages and Their Health Impacts

The burden of disease caused by high sugar-sweetened beverages (SSBs) rose worldwide between 1990 and 2019 [1,2]. In Australia, over half of all children (~60–75%, depending on their age) are estimated to exceed the WHO [3], recommendation that free sugars (“monosaccharides and disaccharides added to foods and beverages by the manufacturer, cook or consumer and sugars naturally present in honey, syrups, fruit juices and fruit juice concentrates” [4]) should not exceed 10% of a person’s total energy intake [5]. Sugar-sweetened beverages are the greatest source of added sugar (21.5%) in the diets of Australian children aged 4–18 years [5]. Sugar-sweetened beverages (SSBs) are defined by the World Health Organization [4] as “all types of beverages containing free sugars, and these include carbonated or non-carbonated soft drinks, fruit/vegetable juices and drinks, liquid and powder concentrates, flavoured water, energy and sports drinks, ready-to-drink tea, ready-to-drink coffee and flavoured milk drinks”.

SSBs are associated with a significant increase in non-communicable diseases, overweight, and obesity in both adults and children across the globe [6,7]. Excess SSB consumption during childhood is associated with a range of health problems and, in particular, unhealthy weight gain [8]. Concerningly, two in five Australian children aged 2–17 years consume SSBs at least once per week and one in fourteen consume SSBs daily [9]. With 25% of Australian children between two and 17 classified as living with overweight or obesity [10], intervening to curb Australian children’s SSB consumption is essential to their current and future health, particularly given that these ill health impacts can carry into adulthood [11].

### 1.2. Public Perceptions of Sugar-Sweetened Beverages

There is evidence of some school-based education and public health mass media campaigns increasing awareness of negative health effects of SSBs and promoting decreased SSB consumption [12,13,14], even when their media presence is vastly outweighed by competing commercial product advertising promoting SSB consumption [15]. As public awareness about the health impacts of SSBs grows, it is important to understand which drinks are perceived to be suitable alternatives. Drinks marketed with a ‘health halo’ [16,17], such as 100% fruit juice and flavoured milks, may seem like equally appealing but healthier alternatives. However, the high-sugar content and health risks associated with these ‘healthy’ alternatives are comparable to soft drinks [18,19,20,21]. Intake of soft drinks alternatives (e.g., 100% fruit juice) in volumes greater than recommended [3] may occur due to their perceived healthiness [22,23].

### 1.3. Parental Perceptions and Provision of Sugar-Sweetened Beverages

Research has shown that primary caregivers tend to exert the most influence on their child’s SSB intake, both in terms of accessibility in the home [24] and modelling behaviour through their own intake [25,26]. Parents are more likely to provide their children with what they perceive as healthier drink alternatives [27]. It is therefore critical to understand how parents perceive these sugary drinks with a health halo, which they may perceive as healthier compared to other sugary drinks.

In addition to understanding whether fruit juice and fruit drinks (drinks containing fruit juice and added sugar) may be perceived as acceptable to consume as healthier drink options, there is also a question as to whether these drinks are substituted for whole fruits [28,29]. Considering that Australia has the highest rate of fruit juice consumption worldwide [30], further research is needed to understand whether this is impacting whole fruit consumption. Another factor which may have affected household sugary drink consumption is the COVID-19 pandemic, which led to increased SSB consumption among children in the US during COVID-19 due to its accessibility in the home, disruption of routine, and as a way to counter boredom [31].

### 1.4. Aims and Objectives

To date, there is limited work examining how perceptions of SSBs relate to consumption amongst young Australian children and their parents [21,27,32]. Most research exploring associations between psycho-social factors and SSB intake has focused on older children and teenagers [33].

The aim of the current study was to examine the consumption and perceptions (relating to healthiness, tastiness, and cost) of various beverages, including soft drinks and beverages that might potentially be considered ‘healthier’ alternatives, among both parents and children. We hypothesised that perceived healthiness among parents and children, as well as parents’ own consumption, would be strong factors associated with child consumption across all drinks. We also examined the relationship between children’s whole fruit consumption and consumption of 100% fruit juice and fruit drinks. Finally, we asked parents and children about the impact of the COVID-19 pandemic on their consumption of different beverages, with the prediction that a slight increase in SSB consumption would be seen as a result of the pandemic.

## 2. Materials and Methods

### 2.1. Participants

Ethical approval was granted by Curtin University Human Research Ethics Committee (HRE2021-0693). Participants were recruited via PureProfile (Sydney, Australia), an online web panel provider, from late March 2023 to late April 2023. Participants had to reside in Australia, be the parent of a child between 4 and 11 years old who was willing to complete the online survey with them, and be the primary food provider for their child. In accordance with the PureProfile payment protocols, participants were paid USD 9 for their time in completing the 15 min survey.

### 2.2. Procedure

Participants first read an information sheet and then provided informed consent. All participants answered three bot detection/attention check questions—(i) a CAPTCHA, (ii) a question requiring participants to select “strongly agree”, and (iii) a question asking participants to type the name of their favourite animal. If any of these questions were answered incorrectly, participants were directed to the end of the survey, and their responses were removed from the dataset prior to analysis. Participants then completed questions assessing their eligibility to complete the survey. If they answered “no” to any questions, they were directed to the end of the survey.

Eligible participants then continued to the main survey, which began with demographic questions. Parents were told that of the following questions, half would be for them to complete in relation to themselves (labelled “Parent Questions”), while the other half would be for their child to complete with parental assistance (labelled “Child Questions”). Participants were asked to estimate their fruit consumption before continuing to complete a series of questions related to their consumption of various beverages, and their perceptions of these beverages. Participants then had to indicate if their consumption of these beverages had changed due to COVID-19. Finally, participants were asked their height and weight. At the end of the survey, participants were debriefed and thanked, and redirected to the PureProfile website to receive payment for their time in completing the survey. As many of the questions were created for the purpose of this survey, an initial version was pilot-tested with four parent–child dyads using a read-aloud methodology [34]. Recordings from these dyads were transcribed and summarised by external coders. Changes were made to the survey based on feedback from both the parents and children.

### 2.3. Measures

The beverages included in this survey were (1) 100% fruit juice, (2) fruit drinks, (3) flavoured milk, (4) plain milk, (5) soft drinks, (6) beverages containing non-nutritive sweeteners (referred to hereafter as NNS beverages), and (7) tap water. For each type of beverage, participants were first asked about their consumption in a typical week and then about their perceptions of the beverage type. Both parent and child completed the same set of questions for the same set of beverage types; however, the wording was slightly modified for some of the perception questions directed at children to aid comprehension, particularly for the younger age groups.

*Demographics.* At the start of the survey, parents were asked their age, gender identity, highest level of education, age of each of their children, age of the child completing the survey with them, gender identity of the child completing the survey with them, and whether they are the main grocery buyer in their household. Parents also reported their postcode, from which socio-economic status was determined using the Index of Relative Socio-economic Advantage and Disadvantage [35]. At the end of the survey, parents were also asked to report their height and weight and their child’s height and weight.

*Fruit consumption.* Parents and children were asked: “How many serves of fruit (including fresh, frozen, and tinned fruit) do you usually eat each day? (A ‘serve’ = 1 medium piece or 2 small pieces of fruit or 1 cup of diced fruit)”. Participants could choose from four response options: “1 serve”, “2 serves”, “3 serves or more”, and “I don’t eat fruit” [36].

*Beverage consumption.* Participants firstly selected from three response options: (1) 250 mL (one cup) or less per week, (2) more than 250 mL per week, (3) I never consume this type of drink. If participants did not select “more than 250 mL per week”, they skipped ahead to the perception questions for that beverage type. If participants selected “more than 250 mL per week”, then they were asked to indicate how many servings of that type of beverage they consumed per week. The serving sizes shown varied according to common serving/packaging sizes for each beverage category (see Appendix A in the Appendix A). For instance, for 100% fruit juice, participants were asked to report the number of servings of 250 mL, 350 mL, 1 L, and 2 L they consumed each week. Total consumption for each beverage type was calculated by multiplying the number of servings reported for each serving size by its volume in millilitres and summing this across each of the serving sizes shown.

*Perceptions of beverage types.* Participants had the option to indicate if they did not know enough about the beverage type to rate it and thus skip the perception questions. If they did not check this box, they were asked to indicate, on a 10-point bipolar rating scale, the extent to which they thought the beverage was (1) healthy/unhealthy, (2) not tasty/tasty, (3) not good for kids/good for kids, (4) cost a lot of money/does not cost much money, (4) high in sugar/low in sugar, and (5) okay to have every day/not okay to have every day. Ratings at the lower and higher end of the 10-point scale indicated stronger agreement with the associated measure (e.g., healthy or unhealthy) while ratings in the middle of the scale indicated a more neutral position.

*Influence of COVID-19 on beverage consumption.* Parents were asked “Do you think your consumption of any type of drink has increased since the first COVID-19 lockdown you experienced?” to which they could choose “yes” or “no”. If participants selected “yes”, they were then asked to select which beverages they have consumed more of from a list of the beverages included in the survey or “other” and specify a different beverage. Parents filled out the same COVID-19 related consumption question for their child.

### 2.4. Data Analysis

Data were analysed using SPSS statistics v29 and R Studio v4.3.3. Data were screened for incomplete responses, with *n* = 99 cases removed due to either dropping out of the survey early, or not having completed the survey sufficiently. The final sample consisted of *N* = 1611 participants.

Using the National Health and Medical Research Council [37] recommended healthy fluid limits and researcher judgment, a total volume of liquid consumption was created to classify participants as error outliers. Weekly consumption of 140 L or 70 L were applied as the cut-off maximum for parents and children, respectively. This resulted in 8 participants removed from the dataset. No minimum cut-off was applied, as there were drink types not measured in our study (e.g., bottled or filtered water, sports drinks etc.).

Winsorization was applied to the remaining error outliers (where participants reported a safe but still unrealistically high level of fluid consumption) to reduce bias to the mean and results during analysis. All consumption data above the 95th percentile was transformed down to the 95th percentile [38]. For example, if child’s total volume for flavoured milk was 1900 mL, and the 95th percentile cap for this drink type was 1800 mL, their data was transformed to 1800 mL for this drink type. Cut-offs for each drink can be found in Appendix A of the Appendix A.

Descriptive statistics (means, standard deviations, and ranges) were used to capture total consumption of each beverage type in millilitres per week, and differences in perceptions of each beverage type between parents and children. A series of paired samples *t*-tests were then run to test for significant differences (with a Bonferroni correction applied and effect sizes reported). This was repeated within the 4–7 and 8–11 child age groups. Demographic variables were also tested for associations with consumption of each beverage type. Given that the consumption variables were strongly positively skewed (on account of the high proportion of those reporting no consumption in each beverage type), Kendall’s Tau was used to assess associations with age, BMI, education, and socioeconomic status with both parent and child consumption [39]. For similar reasons, a series of zero-inflated regression models were constructed to assess the degree by which perceptual factors (e.g., tastiness, healthiness) of each beverage type related to consumption of the beverage. This method specifies two components: a binary component, which predicts the likelihood of observing a zero outcome (in this case 0 mL consumed per week), and a secondary component, which assumes a negative binomial distribution, to predict the level of consumption for non-zero cases [40]. These relationships were tested for attenuation after the addition of associated demographic variables. Kendall’s Tau was used to assess the relationship between whole fruit consumption and fruit juice/fruit drink consumption among children and parents. Finally, descriptive statistics (means, standard deviations, frequencies) were used to determine whether participants felt that their beverage consumption patterns changed after COVID-19.

## 3. Results

### 3.1. Sample Characteristics

The mean age of parents in this study was 37.17 years (*SD* = 7.18 years), and the mean BMI was 27.02 (*SD* = 6.08). Among the sample, 1057 parents (65.7%) identified as female. The mean age of children was 7.27 years (*SD* = 2.38 years), and 755 (46.9%) identified as female. Table 1 shows other participant characteristics.

### 3.2. Beverage Consumption

As shown in Table 2, tap water, plain milk, and NNS beverages were the most consumed beverages by parents, while flavoured milk, 100% fruit juice, and fruit drinks were the least consumed. Tap water, plain milk, and 100% fruit juice were the most consumed beverages by children on average, while NNS beverages, soft drinks, and flavoured milk were the least consumed.

### 3.3. Beverage Perceptions

Figure 1 shows the mean child perceptions for each beverage. Soft drinks were perceived by children as the least healthy drinks (*M* = 2.94) and highest in sugar compared to all other drinks (*M* = 3.03). Tap water was perceived to be the least tasty drink (*M* = 6.02). Soft drinks and NNS beverages were perceived as the beverages that were least good for kids (*M_soft drink_* = 3.67, *M_NNS beverage_* = 4.6) and the least okay to drink every day (*M_soft drink_* = 7.46, *M_NNS beverage_* = 6.81) compared to the other drinks. Compared to the other drinks, tap water was perceived as the drink costing the least amount of money (*M* = 8.44).

The results from the series of paired samples t-tests comparing beverage perceptions between parents and children revealed that there were some significant differences in the ratings of beverages, but for the most part, these differences were not large. As multiple comparisons were performed, the strength of the relationship was interpreted by the effect size (*d*) rather than the significance value (*p*). Children of all ages perceived 100% fruit juice as moderately (*d* > 0.4) healthier, good for kids, and okay to have every day, compared to parents. These effects were stronger for fruit drinks (*d* = 0.39–0.58) and slightly more so for flavoured milk (*d* = 0.42–0.62). Perceptions of plain milk, soft drinks, NNS beverages, and (especially) tap water were generally not strongly divergent across parents and children (of all ages), although there were still some smaller effects present (e.g., *d* = 0.12–0.34). The exception was that children (particularly young children, *d* = 0.49) had a stronger perception that soft drinks were good for kids compared to their parents, and that it was okay to have them every day (*d* = 0.38). NNS beverages were rated as being more acceptable to have every day by older children (*d* = 0.17) when compared to their parents, but not by younger children. However, younger children rated NNS beverages as higher in sugar compared to parents (*d* = 0.27), whereas older children’s ratings did not differ from their parents.

### 3.4. Factors Related to Beverage Consumption

Table 3 reports the associations of each demographic variable with both parent and child drink consumption. Higher parent BMI scores predicted greater soft drink and NNS beverage consumption in parents. Education was significantly inversely related to parent soft drink consumption and was weakly positively related to 100% fruit juice consumption (but not statistically significant after correcting for multiple hypothesis comparisons). Higher SES was weakly but significantly related to NNS beverage consumption in parents.

Child age was a predictor of consumption for both soft drinks and NNS beverages, with older children consuming more of each drink type. This was true for fruit drink as well, but the effect was weaker. After Bonferroni corrections, only soft drink consumption was related to weight, such that greater consumption corresponded with higher parent-reported child weights. Importantly, higher parent education was also significantly but weakly related to greater child consumption of both 100% fruit juice and plain milk. SES was not related to children’s consumption of any beverage type, nor was child gender.

The series of zero-inflated regression analyses were conducted examining the influence of child and parent perceptual factors, as well as parent consumption, on child consumption (with coefficients shown in Table 4). No relationships between predictors and behaviour changed substantially after controlling for relevant demographic variables. Nonetheless, the results shown are from models which include demographic variables where relevant (i.e., if they were found to be related to the outcome in the section prior), and these models are denoted with superscripts.

For the most part, child perceptions were not predictive of the quantity of beverages consumed by the child (the negative binomial component). For detailed reporting of models that included only child perceptions, please refer to Appendix A of the Appendix A. When the parent variables (perceptions and consumption) were added into the model (Table 4), any significant child perception–consumption relationships became attenuated and non-significant after Bonferroni corrections, with only parent consumption significantly predicting child consumption. For each extra litre of the same beverage type that was consumed by parents, the predicted increase in child consumption ranged from a 6.29% increase for soft drinks to a 17.5% increase for 100% fruit juice.

The inspection of the logistic components (in Appendix A of the Appendix A) revealed the importance of several child perceptions in predicting whether a child consumed that beverage or not. However, as with the negative binomial component, when parent variables were added into the model (see Table 4), parent consumption consistently predicted child consumption across all drink categories, such that for every extra litre of the respective beverage type consumed by parents, the odds of children reporting 0 mL of consumption decreased, with estimates ranging from 17.72% for plain milk to 54.52% for flavoured milk. Only one parent perception was identified as important in predicting child consumption, with increased parent ratings of flavoured milk as being good for kids, corresponding to 16.89% reduced odds of children consuming 0 mL in a typical week.

Given the strong predictive power of parent consumption on child consumption, zero-inflated regression analyses were conducted to examine the influence of parents’ perceptions on their own consumption (Table 5). The strongest predictor of consumption across both parents and children was their perception of whether a beverage was okay to have every day. The stronger participants felt that a particular beverage was not okay to have every day, the less likely they were to drink it. Additionally, tastiness was a strong predictor (stronger than perceived healthiness) of whether parents consumed soft drinks, NNS beverages, or flavoured milk and how much they consumed.

### 3.5. Relationship between Whole Fruit Consumption and Fruit Juice/Fruit Drink Consumption

Parent fruit consumption was moderately associated with child fruit consumption, τ = 0.395, *p* < 0.001. Parents’ fruit consumption was also weakly related to children’s 100% fruit juice consumption, τ = 0.077, *p* < 0.001, and plain milk consumption, τ = 0.085, *p* < 0.001. Children’s fruit consumption was only related to plain milk consumption, τ = 0.088, *p* < 0.001, and tap water consumption, τ = 0.128, *p* < 0.001, but not to either type of fruit beverage consumption.

### 3.6. Influence of COVID-19 on Beverage Consumption

Of the 559 (34.7%) parents who reported an increase in their consumption of any drink since the first COVID-19 lockdown they experienced, the most frequent drink types were tap water (39%), plain milk (36.5%), and, equally, 100% fruit juice and soft drinks (34.7%). Similarly, 424 (26.3%) parents indicated an increase in their child’s consumption of any drink since the first COVID-19 lockdown they experienced, with the most frequent drink types being plain milk (48.3%), 100% fruit juice (45%), and tap water (37.7%), closely followed by flavoured milk (34.7%).

## 4. Discussion

The aim of this study was to better understand the perceptions and consumption patterns across a range of drink types, with a particular focus on drinks marketed as ‘healthy’ alternatives (e.g., 100% fruit juice), among young children and their parents. Specifically, this study sought to understand how parents’ and children’s perceptions of different drinks, and parents’ consumption behaviours, relate to children’s consumption. Additionally, this study explored the relationship between the consumption of whole fruit, 100% fruit juice, and fruit drinks, as well as the potential impact of the COVID-19 pandemic on children’s intake of various beverages. Given that this topic has not been well studied in young children, our findings shed light across many areas.

Despite their comparable sugar content to soft drinks, children perceived 100% fruit juice, fruit drinks, and flavoured milk to be substantially healthier, lower in sugar and better for kids. Children perceived 100% fruit juice and NNS drinks to have a similar sugar content, considering it to be lower than that of all other drinks included in this study, except for tap water and plain milk. Fruit drinks and flavoured milk were also perceived to have a similar sugar content, which was lower than that of soft drinks. Although some child perceptions predicted child consumption, these relationships were no longer significant once parent consumption was accounted for. Parents’ consumption was the most significant predictor of child consumption across all drinks. This finding emphasizes the pivotal role parents appear to play, both as role models and gatekeepers of their children’s sugary drink consumption [41,42]. This is consistent with past studies showing the influence parents have on their children’s diet [43,44] but is one of only a few studies to show this across a range of beverages and among a sample of young children. Uniquely, this study shows that children’s sugary drink consumption bears a stronger association with parental modelling of this behaviour than children’s personal perceptions of sugary drinks.

When comparing children’s and parents’ perceptions of the different beverages, children generally perceived all beverages more positively than parents (in terms of healthiness, tastiness, cost, whether the drink was good for kids, acceptable to have every day, and low in sugar). However, where this difference was substantially divergent was in children’s positive perceptions of the health attributes of 100% fruit juice, fruit drinks, flavoured milk, and soft drinks. This divergence was particularly prominent for the younger children in our sample (i.e., those aged 4–7 years), suggesting that younger children hold perceptions of certain drinks that are not necessarily shared by their parents. This finding diverges from past research, demonstrating the importance of parental perceptions in guiding child perceptions [23]. However, it is illuminating, given the lack of studies in children this young (4–7 years old). It is possible that these children’s views on sugary drinks reflect the food marketing environment, since Australian children are heavily exposed to unhealthy food and sugary drink advertising across a range of media and settings [45], which is known to influence children’s dietary attitudes, beliefs, and behaviour [46], with younger children thought to be more vulnerable to being misled by such advertising due to their greater cognitive and developmental immaturity [47]. Further research is needed to better understand the relative influence of various socio-cultural factors on the development of young children’s perceptions of different beverages. However, accumulating evidence indicates that policies to protect children from exposure to unhealthy food and drink advertising is a crucial component of comprehensive efforts to encourage and support healthy eating in childhood [48], which could also empower parents and schools to exert more positive influence on children’s beliefs and behaviour regarding sugary drinks. School-based education campaigns, which have been shown to successfully reduce SSB consumption [12,13,14], could address a wider range of sugary drinks to counter erroneous perceptions regarding their sugar levels and nutritional attributes, especially among younger children.

There was some evidence to suggest that the concept of NNS beverages was not fully grasped by younger children, as they tended to rate these beverages as having higher sugar than their parents, whereas older children did not. In contrast, older children rated NNS beverages as being more acceptable to have every day. Our results are in line with previous studies, showing that consumption of soft drinks and NNS beverages increases with child age [49]. Studies assessing longitudinal trends in sugary drink consumption show that children’s early experience of sweet tastes is a strong predictor of their consumption behaviours in the future [11,28]. Thus, it is important to reduce exposure to sugary drinks in early childhood for short- and long-term health outcomes [9,18].

Our results also showed that there was no relationship between 100% fruit juice or fruit drink consumption and whole fruit consumption, suggesting that children’s juice intake was not displacing fruit consumption. This is in line with past research [28]. Additionally, for most of the sample, there did not appear to be any self-reported long-term influence (four years later) of COVID-19 on increased consumption of sugary drinks.

### Limitations

The present findings relied on self-reports from one time point. While the survey was designed and pilot-tested to increase accessibility (particularly for young children), with instructions to parents on how to assist their child, use of simplified language for children and inclusion of visual aids, it is still possible that the parents and children (particularly the youngest children) did not accurately recall their weekly consumption of beverages (including how this changed with the pandemic) or their height and weight. Data were collected at one time point to minimize participant burden and attrition, but this means that we can only report on associations rather than causal relationships.

## 5. Conclusions

It is important to recognise the role parents play in shaping their children’s perceptions and consumption behaviours towards sugary drinks, and as such, interventions should prioritise educating parents about the nutritional content of different drinks, particularly those that are often seen as healthier options, as well as the impact of their own consumption on their child/ren’s consumption [28]. This could be achieved through public health education campaigns and initiatives targeted at adults, such as LiveLighter [12] and Rethink Sugary Drinks [50], as well as parent education workshops, such as Packed with Goodness [51], in which parents are given practical advice on providing healthy foods and drinks to children, including information on the nutritional content of commonly purchased drinks.

A secondary strategy would be to educate children (particularly young children) on the healthiness of various drink options. Educational campaigns and public health marketing efforts could inform both parents and children about the sugar content and health risks of supposedly healthy alternative beverages [12,13].

Future research should investigate labelling practices that are associated with marketed healthy drink alternatives to identify potential areas for improvement (such as providing unbiased information to prevent health halo effects). These findings provide useful information to public health agencies for their program development and mass media campaign planning.

## Figures and Tables

**Figure 1 nutrients-16-03320-f001:**
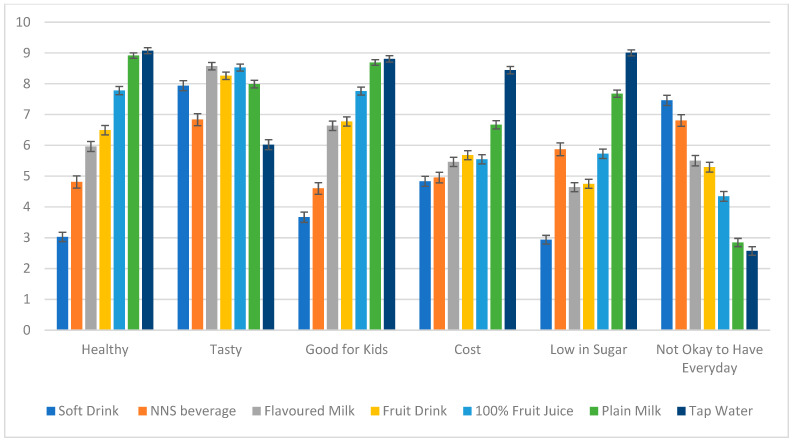
Mean child perceptions for each drink type with 95% confidence intervals. Note. Some children selected that they did not know the drink type (besides tap water), and thus did not provide their perceptions: 100% fruit juice (*n* = 306), fruit drinks (*n* = 334), flavoured milk (*n* = 362), plain milk (*n* = 133), soft drinks (*n* = 479), NNS drinks (*n* = 660).

**Table 1 nutrients-16-03320-t001:** Sample descriptives.

Variable	Response Options	*n* (%)
SES Quintile		
	1	202 (12.5)
	2	254 (15.8)
	3	356 (22.1)
	4	365 (22.7)
	5	414 (25.7)
Highest Education Level		
	Primary	9 (0.6)
	Secondary	293 (18.2)
	TAFE/Technical College	475 (29.5)
	University, undergraduate	545 (33.8)
	University, postgraduate	288 (17.9)
Main Grocery Buyer		
	Mainly me	1386 (86.0)
	Mainly someone else	17 (1.1)
	Equally shared	207 (12.8)
Parent Gender		
	Male	551 (34.2)
	Female	1057 (65.6)
	Non-binary	0 (0)
	Prefer not to say	2 (0.1)
Child Gender		
	Male	851 (52.8)
	Female	755 (46.9)
	Non-binary	0 (0)
	Prefer not to say	4 (0.2)

**Table 2 nutrients-16-03320-t002:** Weekly consumption of each beverage type for parents and children.

Consumer	Drink Type	Consumption Volume *n* (%)			Mean Consumption in mL (*SD*)
		Never Consume	250 mL or Less	More than 250 mL	
Parent	100% Fruit Juice	424 (26.3)	675 (41.9)	512 (31.8)	522 (1000)
	Fruit Drinks	545 (33.8)	652 (40.5)	414 (25.7)	492 (1056)
	Flavoured Milk	699 (43.4)	574 (35.6)	338 (21.0)	291 (678)
	Plain Milk	199 (12.4)	324 (20.1)	1088 (67.5)	1863 (2431)
	Soft Drinks	430 (26.7)	648 (40.2)	533 (33.1)	642 (1244)
	NNS Beverage	539 (33.5)	520 (32.3)	550 (34.1)	701 (1275)
	Tap Water	111 (6.9)	92 (5.7)	1408 (87.4)	4144 (4183)
Child	100% Fruit Juice	391 (24.3)	679 (42.1)	539 (33.5)	419 (713)
	Fruit Drinks	495 (30.7)	670 (41.6)	445 (27.6)	329 (610)
	Flavoured Milk	547 (34.0)	698 (43.3)	364 (22.6)	230 (500)
	Plain Milk	175 (10.9)	296 (18.4)	1139 (70.7)	1458 (1592)
	Soft Drinks	800 (49.7)	554 (34.4)	257 (16.0)	142 (357)
	NNS Beverage	1016 (63.1)	420 (26.1)	174 (10.8)	98 (304)
	Tap Water	90 (5.6)	106 (6.6)	1415 (87.8)	4440 (3354)

**Table 3 nutrients-16-03320-t003:** Demographic associations (Kendall’s Tau) with parent and child consumption.

Consumer	Demographic Variable	Beverage Type	*τ*	*p*
Parent	Age	100% Fruit Juice	0.009	0.628
		Fruit Drinks	−0.024	0.225
		Flavoured Milk	−0.032	0.102
		Plain Milk	0.007	0.683
		Soft Drinks	−0.044	0.022
		NNS Beverages	0.026	0.173
		Tap Water	−0.019	0.278
	BMI	100% Fruit Juice	−0.053	0.006
		Fruit Drinks	−0.018	0.353
		Flavoured Milk	−0.013	0.497
		Plain Milk	0.007	0.711
		Soft Drinks	0.112	<0.001
		NNS Beverages	0.137	<0.001
		Tap Water	0.033	0.059
	Education	100% Fruit Juice	0.066	0.002
		Fruit Drinks	0.019	0.370
		Flavoured Milk	0.010	0.639
		Plain Milk	0.051	0.011
		Soft Drinks	−0.100	0.001
		NNS Beverages	−0.020	0.338
		Tap Water	0.007	0.731
	SES	100% Fruit Juice	0.038	0.070
		Fruit Drinks	0.018	0.410
		Flavoured Milk	−0.024	0.254
		Plain Milk	0.027	0.171
		Soft Drinks	−0.010	0.628
		NNS Beverages	0.067	0.001
		Tap Water	0.020	0.301
Child	Child Age	100% Fruit Juice	0.036	0.076
		Fruit Drinks	0.078	<0.001
		Flavoured Milk	0.041	0.044
		Plain Milk	−0.009	0.628
		Soft Drinks	0.160	<0.001
		NNS Beverages	0.124	<0.001
		Tap Water	0.022	0.222
	Weight	100% Fruit Juice	0.04	0.303
		Fruit Drinks	0.11	0.005
		Flavoured Milk	0.03	0.524
		Plain Milk	0.04	0.248
		Soft Drinks	0.16	<0.001
		NNS Beverages	0.11	0.009
		Tap Water	−0.01	0.859
	Parent Education	100% Fruit Juice	0.074	<0.001
		Fruit Drinks	−0.019	0.382
		Flavoured Milk	−0.007	0.730
		Plain Milk	0.064	<0.001
		Soft Drinks	0.002	0.923
		NNS Beverages	−0.011	0.608
		Tap Water	−0.030	0.112
	SES	100% Fruit Juice	0.013	0.525
		Fruit Drink	0.020	0.344
		Flavoured Milk	0.003	0.905
		Plain Milk	0.028	0.149
		Soft Drink	0.006	0.766
		NNS Beverages	−0.023	0.300
		Tap Water	0.003	0.878

**Table 4 nutrients-16-03320-t004:** Coefficients from zero-inflated regression analyses predicting child consumption of each beverage type with child and parent perceptions and parent consumption.

		Negative Binomial			Logistic		
Beverage Type	Predictor	Coefficient	Std. Error	*p*	Coefficient	Std. Error	*p*
100% Fruit Juice ^a^	Healthy	−0.023	0.015	0.123	0.039	0.043	0.359
	Tasty	0.033	0.019	0.075	−0.225	0.046	<0.001
	Good For Kids	0.046	0.016	0.005	0.002	0.044	0.964
	Cheap	−0.020	0.009	0.032	−0.025	0.027	0.353
	Low in Sugar	0.012	0.01	0.219	0.005	0.028	0.85
	Not Okay to Have Every day	−0.004	0.01	0.719	0.156	0.029	<0.001
	Healthy (Parent)	−0.010	0.017	0.551	0.048	0.047	0.306
	Tasty (Parent)	−0.003	0.014	0.859	0.078	0.044	0.073
	Good For Kids (Parent)	0.024	0.019	0.203	−0.137	0.051	0.007
	Cheap (Parent)	0.015	0.01	0.123	−0.022	0.029	0.45
	Low in Sugar (Parent)	0.012	0.011	0.283	−0.032	0.033	0.333
	Not Okay to Have Every day (Parent)	0.005	0.01	0.623	0.087	0.031	0.005
	Parent Consumption	0.162	0.019	<0.001	−0.617	0.07	<0.001
Fruit Drink ^b^	Healthy	0.004	0.015	0.800	0.016	0.043	0.716
	Tasty	0.008	0.016	0.647	−0.143	0.044	<0.001
	Good For Kids	0.001	0.016	0.926	−0.065	0.047	0.166
	Cheap	0.006	0.01	0.561	0.046	0.031	0.135
	Low in Sugar	−0.002	0.011	0.882	−0.041	0.033	0.217
	Not Okay to Have Every day	−0.028	0.010	0.008	0.149	0.03	<0.001
	Healthy (Parent)	−0.018	0.015	0.235	−0.008	0.046	0.87
	Tasty (Parent)	−0.018	0.013	0.182	0.012	0.038	0.75
	Good For Kids (Parent)	0.036	0.016	0.025	−0.007	0.048	0.887
	Cheap (Parent)	0.003	0.012	0.799	−0.032	0.035	0.349
	Low in Sugar (Parent)	0.030	0.013	0.019	−0.066	0.04	0.097
	Not Okay to Have Every day (Parent)	0.017	0.012	0.141	0.063	0.036	0.076
	Parent Consumption	0.111	0.02	<0.001	−0.383	0.065	<0.001
Flavoured Milk	Healthy	−0.015	0.016	0.377	−0.119	0.039	0.002
	Tasty	0.018	0.021	0.387	−0.072	0.048	0.132
	Good For Kids	0.009	0.018	0.614	−0.045	0.045	0.317
	Cheap	−0.001	0.012	0.923	0.014	0.032	0.663
	Low in Sugar	−0.016	0.014	0.248	0.035	0.036	0.328
	Not Okay to Have Every day	−0.021	0.012	0.0820	0.087	0.030	0.004
	Healthy (Parent)	0.020	0.019	0.305	0.088	0.047	0.059
	Tasty (Parent)	−0.005	0.02	0.798	0.037	0.045	0.406
	Good For Kids (Parent)	−0.012	0.02	0.545	−0.185	0.051	<0.001
	Cheap (Parent)	0.025	0.015	.087	0.025	0.039	0.523
	Low in Sugar (Parent)	0.009	0.016	0.564	−0.034	0.042	0.410
	Not Okay to Have Every day (Parent)	0.006	0.015	0.686	−0.006	0.036	0.874
	Parent Consumption	0.158	0.033	<0.001	−0.788	0.096	<0.001
Plain Milk ^a^	Healthy	0.004	0.016	0.814	−0.061	0.052	0.24
	Tasty	0.020	0.010	0.056	−0.307	0.031	<0.001
	Good For Kids	0.009	0.016	0.556	−0.025	0.052	0.631
	Cheap	−0.003	0.007	0.697	0.016	0.03	0.589
	Low in Sugar	0.011	0.009	0.226	−0.003	0.035	0.928
	Not Okay to Have Every day	0.017	0.009	0.059	0.11	0.031	<0.001
	Healthy (Parent)	0.009	0.015	0.555	0.077	0.056	0.172
	Tasty (Parent)	−0.002	0.009	0.783	0.109	0.037	0.003
	Good For Kids (Parent)	−0.004	0.015	0.777	−0.152	0.055	0.006
	Cheap (Parent)	−0.002	0.008	0.759	−0.003	0.033	0.928
	Low in Sugar (Parent)	0.018	0.009	0.053	0.024	0.038	0.53
	Not Okay to Have Every day (Parent)	0.013	0.008	0.122	0.007	0.031	0.822
	Parent Consumption	0.116	0.007	<0.001	−0.195	0.035	<0.001
Soft Drinks ^b^	Healthy	0.019	0.015	0.189	−0.135	0.048	0.005
	Tasty	0.041	0.020	0.041	−0.154	0.046	<0.001
	Good For Kids	−0.003	0.016	0.865	−0.009	0.047	0.848
	Cheap	0.005	0.012	0.673	−0.044	0.036	0.216
	Low in Sugar	0.009	0.016	0.583	0.023	0.05	0.649
	Not Okay to Have Every day	−0.015	0.012	0.208	0.124	0.036	<0.001
	Healthy (Parent)	0.003	0.023	0.904	−0.019	0.07	0.790
	Tasty (Parent)	0.004	0.018	0.809	−0.023	0.048	0.636
	Good For Kids (Parent)	0.025	0.021	0.232	−0.015	0.071	0.831
	Cheap (Parent)	−0.003	0.012	0.802	0.002	0.037	0.958
	Low in Sugar (Parent)	−0.018	0.015	0.204	−0.041	0.05	0.412
	Not Okay to Have Every day (Parent)	0.012	0.013	0.383	0.001	0.045	0.977
	Parent Consumption	0.061	0.017	0.001	−0.38	0.06	<0.001
NNS Beverages ^c^	Parent Consumption	0.016	0.020	0.425	−0.095	0.051	0.060
	Tasty	0.028	0.019	0.139	−0.149	0.048	0.002
	Good For Kids	−0.028	0.02	0.166	−0.047	0.056	0.400
	Cheap	−0.003	0.015	0.822	−0.109	0.045	0.015
	Low in Sugar	−0.011	0.015	0.455	−0.008	0.043	0.853
	Not Okay to Have Every day	−0.013	0.014	0.347	0.075	0.041	0.065
	Healthy (Parent)	0.015	0.020	0.455	0.042	0.059	0.481
	Tasty (Parent)	−0.021	0.020	0.289	−0.023	0.051	0.649
	Good For Kids (Parent)	0.011	0.02	0.561	−0.077	0.060	0.196
	Cheap (Parent)	0.006	0.015	0.698	−0.056	0.047	0.233
	Low in Sugar (Parent)	−0.006	0.015	0.699	0.078	0.043	0.068
	Not Okay to Have Every day (Parent)	−0.004	0.015	0.769	0.021	0.047	0.648
	Parent Consumption	0.081	0.021	<0.001	−0.366	0.067	<0.001
Tap Water	Healthy	0.050	0.016	0.002	−0.186	0.063	0.003
	Tasty	0.013	0.005	0.009	−0.001	0.037	0.989
	Good For Kids	0.015	0.013	0.251	−0.131	0.066	0.047
	Cheap	−0.003	0.007	0.621	0.075	0.052	0.152
	Low in Sugar	0.007	0.010	0.473	0.000	0.057	0.999
	Not Okay to Have Every day	−0.009	0.007	0.179	0.114	0.037	0.002
	Healthy (Parent)	−0.019	0.013	0.154	−0.051	0.066	0.444
	Tasty (Parent)	−0.015	0.006	0.010	0.011	0.041	0.778
	Good For Kids (Parent)	0.028	0.014	0.039	−0.165	0.068	0.015
	Cheap (Parent)	0.007	0.007	0.29	0.053	0.047	0.261
	Low in Sugar (Parent)	−0.001	0.009	0.909	−0.062	0.054	0.251
	Not Okay to Have Every day (Parent)	−0.013	0.007	0.064	−0.006	0.038	0.878
	Parent Consumption	0.089	0.003	<0.002	−0.338	0.046	<0.001

^a^ Controlled for parent education. ^b^ Controlled for child age and weight. ^c^ Controlled for child age.

**Table 5 nutrients-16-03320-t005:** Coefficients from zero-inflated regression analyses predicting parent consumption of each beverage type based on perceptions.

		Negative Binomial			Logistic		
Beverage Type	Predictor	Coefficient	Std. Error	*p*	Coefficient	Std. Error	*p*
100% Fruit Juice ^a^	Healthy	0.018	0.020	0.386	−0.118	0.041	0.004
	Tasty	0.000	0.019	0.988	−0.003	0.038	0.944
	Good For Kids	−0.007	0.022	0.759	−0.07	0.043	0.109
	Cheap	0.032	0.014	0.002	−0.079	0.025	0.002
	Low in Sugar	0.030	0.015	0.037	−0.072	0.028	0.010
	Not Okay to Have Every day	0.002	0.013	0.896	0.115	0.026	<0.001
Fruit Drinks ^b^	Healthy	−0.010	0.021	0.646	−0.140	0.04	<0.001
	Tasty	0.013	0.021	0.562	−0.146	0.040	<0.001
	Good For Kids	0.026	0.024	0.273	−0.099	0.045	0.028
	Cheap	0.022	0.016	0.172	−0.039	0.034	0.247
	Low in Sugar	0.013	0.017	0.442	−0.093	0.035	0.008
	Not Okay to Have Every day	0.017	0.015	0.251	0.155	0.032	<0.001
Flavoured Milk ^b^	Healthy	0.008	0.023	0.742	−0.208	0.044	<0.001
	Tasty	0.063	0.02	<0.001	−0.065	0.04	0.109
	Good For Kids	−0.001	0.025	0.966	−0.034	0.048	0.483
	Cheap	−0.005	0.02	0.781	−0.036	0.037	0.329
	Low in Sugar	0.028	0.018	0.120	−0.037	0.038	0.320
	Not Okay to Have Every day	0.013	0.015	0.379	0.164	0.031	<0.001
Plain Milk ^b^	Healthy	0.035	0.02	0.090	−0.042	0.047	0.377
	Tasty	0.101	0.015	<0.001	−0.263	0.028	<0.001
	Good For Kids	−0.038	0.02	0.055	−0.05	0.047	0.29
	Cheap	−0.006	0.011	0.544	0.051	0.028	0.070
	Low in Sugar	0.032	0.012	0.009	0.019	0.032	0.546
	Not Okay to Have Every day	0.04	0.009	<0.001	0.051	0.024	0.035
Soft Drinks ^c^	Healthy	−0.015	0.024	0.53	−0.128	0.049	0.009
	Tasty	0.06	0.019	0.001	−0.273	0.035	<0.001
	Good For Kids	0.041	0.027	0.122	0.015	0.051	0.774
	Cheap	0.001	0.014	0.949	0.04	0.025	0.109
	Low in Sugar	0.020	0.018	0.278	−0.055	0.037	0.136
	Not Okay to Have Every day	−0.04	0.016	0.012	0.174	0.032	<0.001
NNS Beverages ^d^	Healthy	0.040	0.016	0.013	−0.076	0.037	0.041
	Tasty	0.044	0.015	0.004	−0.325	0.031	<0.001
	Good For Kids	−0.016	0.016	0.319	0.099	0.038	0.008
	Cheap	−0.002	0.014	0.906	−0.011	0.03	0.711
	Low in Sugar	0.007	0.011	0.518	−0.050	0.022	0.025
	Not Okay to Have Every day	−0.041	0.012	0.001	0.264	0.030	<0.001
Tap Water	Healthy	0.022	0.017	0.199	−0.079	0.047	0.093
	Tasty	0.011	0.007	0.136	−0.048	0.02	0.017
	Good For Kids	0.035	0.018	0.056	−0.153	0.048	0.002
	Cheap	0.005	0.009	0.556	0.052	0.026	0.047
	Low in Sugar	−0.002	0.012	0.836	0.014	0.034	0.682
	Not Okay to Have Every day	−0.034	0.008	<0.001	−0.007	0.024	0.754

Note. ^a^ Controlled for education. ^b^ Controlled for gender. ^c^ Controlled for age, BMI, education, and gender. ^d^ Controlled for BMI and SES. The logistic component coefficients represent the log odds ratio of consuming 0 mL compared to consuming more than 0 mL, whereas the zero-inflated negative binomial component coefficients represent the expected change in the log of the outcome (consumption in mL).

## Data Availability

The original contributions presented in this study are included in the article; further inquiries can be directed to the corresponding author. The data are not publicly available due to ethical reasons.

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
