# Peer review of "Exploring Sugary Drink Consumption and Perceptions among Primary-School-Aged Children and Parents in Australia"

_nutrients, 2024, doi:10.3390/nu16193320_

Round 1

Reviewer 1 Report

Comments and Suggestions for Authors

Thank you for submitting the manuscript "Exploring sugary drink consumption and perceptions among primary-school aged children and parents in Australia" to Nutrients. Overall, the research appears to have been well conducted. However, the discussion is weak and poorly supported by the literature.

 - consider checking the journal's standard citation form.

 - it would be better to rewrite the introduction in just one text and focusing on the justification of the work. Lines#104-109 are unnecessary.

 - 4 years is an unusual age for the participant to self-report their consumption. It is necessary to add a discussion of this limitation in the manuscript.

 - Lines#372-380: these statements and others in the discussion must be supported by literature. Especially this statement, for example, goes against numerous works that indicate that yes, children learn through observation and repetition especially from parental behavior.

Reviewer 2 Report

Comments and Suggestions for Authors

The manuscript tackles a highly relevant and timely issue, examining the consumption of sugar-sweetened beverages (SSBs) among children and parents. Given the global concerns about the health impacts of excessive SSB intake, the findings of this study provide valuable insights that can help shape effective public health strategies. The methodology is solid, and the results are clearly presented. Including both the perspectives of parents and children adds an important layer to the study, though it would be beneficial to further explore how young children, in particular, develop their views on these beverages.

It's particularly noteworthy that even children in the youngest age group were able to provide opinions about the healthiness and other qualities of the drinks. This, however, raises a question about how much these young children truly understand about the nutritional content of what they are consuming. While the study effectively gathers this data, it would be interesting to see a deeper examination of how accurately such young children can evaluate these attributes.

Round 2

Reviewer 1 Report

Comments and Suggestions for Authors

The manuscript has been carefully reviewed, and the authors have satisfactorily addressed all comments and suggestions presented in the first round of review. The changes made demonstrate a commitment to improving the work, resulting in a clearer and more cohesive text. Given the revisions made and the attention to the comments, I consider that the manuscript now meets the necessary criteria for publication.